[Supplementary Material]

**Supplementary information for "Universality and Individuality in neural dynamics across large populations of recurrent networks"**

## A   RNN Architectures

We examined four RNN architectures that exemplify various degrees of complexity and sophistication. Vanilla RNNs have been historically favored by computational neuroscientists [4, 13], while LSTM and GRU networks have been favored by machine learning practitioners due to performance advantages [20]. However, neuroscientists are beginning to utilize gated RNNs as they progress to studying more complex phenomena [56]. Thus, it is of great interest to determine whether similar mechanisms arise across this range of model architectures, or if different models give rise to distinct dynamics and scientific conclusions. These are architectures are summarized below, with $\mathbf{W}$ and $\mathbf{b}$ respectively representing trainable weight matrices and bias parameters. All other vectors ($\mathbf{c}, \mathbf{g}, \mathbf{r}, \mathbf{i}, \mathbf{f}$) represent intermediate quantities; $\sigma()$ represents a pointwise sigmoid nonlinearity; and $f()$ is either the ReLU or tanh nonlineary.

**Vanilla RNN**
$$\mathbf{h}_t = f(\mathbf{W}^{\text{hh}}\mathbf{h}_{t-1} + \mathbf{W}^{\text{hx}}\mathbf{x}_t + \mathbf{b}^{\text{h}}) \tag{1}$$

**Update-Gate RNN (UGRNN; [20])**
$$\mathbf{h}_t = \mathbf{g} \cdot \mathbf{h}_{t-1} + (1 - \mathbf{g}) \cdot \mathbf{c} \qquad \begin{aligned} \mathbf{c} &= f(\mathbf{W}^{\text{ch}}\mathbf{h}_{t-1} + \mathbf{W}^{\text{cx}}\mathbf{x}_t + \mathbf{b}^{\text{c}}) \\ \mathbf{g} &= \sigma(\mathbf{W}^{\text{gh}}\mathbf{h}_{t-1} + \mathbf{W}^{\text{gx}}\mathbf{x}_t + \mathbf{b}^{\text{g}} + b^{fg}) \end{aligned} \tag{2}$$

**Gated Recurrent Unit (GRU; [18])**
$$\mathbf{h}_t = \mathbf{g} \cdot \mathbf{h}_{t-1} + (1 - \mathbf{g}) \cdot \mathbf{c} \qquad \begin{aligned} \mathbf{c} &= f(\mathbf{W}^{\text{ch}}(\mathbf{r} \cdot \mathbf{h}_{t-1}) + \mathbf{W}^{\text{cx}}\mathbf{x}_t + \mathbf{b}^{\text{c}}) \\ \mathbf{g} &= \sigma(\mathbf{W}^{\text{gh}}\mathbf{h}_{t-1} + \mathbf{W}^{\text{gx}}\mathbf{x}_t + \mathbf{b}^{\text{g}} + b^{fg}) \\ \mathbf{r} &= \sigma(\mathbf{W}^{\text{rh}}\mathbf{h}_{t-1} + \mathbf{W}^{\text{rx}}\mathbf{x}_t + \mathbf{b}^{\text{r}}) \end{aligned} \tag{3}$$

**Long-Short-Term-Memory (LSTM; [17])**
$$\mathbf{h}_t = \begin{bmatrix} \mathbf{c}_t \\ \widetilde{\mathbf{h}}_t \end{bmatrix} \qquad \begin{aligned} \widetilde{\mathbf{h}}_t &= f(\mathbf{c}_t) \cdot \sigma(\mathbf{W}^{\text{hh}}\mathbf{h} + \mathbf{W}^{\text{hx}}\mathbf{x} + \mathbf{b}^{\text{h}}) \\ \mathbf{c}_t &= \mathbf{f}_t \cdot \mathbf{c}_{t-1} + \mathbf{i} \cdot \sigma(\mathbf{W}^{\text{ch}}\widetilde{\mathbf{h}}_{t-1} + \mathbf{W}^{\text{cx}}\mathbf{x} + \mathbf{b}^{\text{c}}) \\ \mathbf{i} &= \sigma(\mathbf{W}^{\text{ih}}\mathbf{h} + \mathbf{W}^{\text{ix}}\mathbf{x} + \mathbf{b}^{\text{i}}) \\ \mathbf{f} &= \sigma(\mathbf{W}^{\text{fh}}\mathbf{h} + \mathbf{W}^{\text{fx}}\mathbf{x} + \mathbf{b}^{\text{f}} + b^{fg}) \end{aligned} \tag{4}$$

## B   Non-normal linear dynamical systems analysis

We studied the linearized systems, e.g. $\frac{\partial F(\mathbf{h}^*, \mathbf{x}^*)_i}{\partial h_j^*}$ using the eigenvector decomposition for non-normal matrices, dropping the dependence on $\mathbf{h}^*$ and $\mathbf{x}^*$ for clarity

$$\mathbf{J} = \mathbf{R}\mathbf{\Lambda}\mathbf{L} = \sum_{a=1}^{N} \lambda_a \mathbf{r}_a \boldsymbol{\ell}_a^{\mathsf{T}}, \tag{5}$$

where $\mathbf{L} = \mathbf{R}^{-1}$, the columns of $\mathbf{R}$ (denoted $\mathbf{r}_a$) contain the *right eigenvectors* of $\mathbf{J}^{\text{rec}}$, the rows of $\mathbf{L}$ (denoted $\boldsymbol{\ell}_a^{\mathsf{T}}$) contain the *left eigenvectors* of $\mathbf{J}^{\text{rec}}$, and $\mathbf{\Lambda}$ is a diagonal matrix containing complex-valued eigenvalues, $\lambda_1 > \lambda_2 > \ldots > \lambda_N$, which are sorted based on their magnitude. Note in particular there is no requirement that $\mathbf{R}^{\mathsf{T}}\mathbf{R} = \mathbf{I}$, leading to potentially sophisticated locally linear dynamics.

## C   Network performance

Below, we include a plot of the final performance of all networks retained for analysis in this paper. All networks achieve low error for their respective tasks.

Figure 5: Histogram of final performance (mean squared error) across all networks used for the analyses in this paper.

## D  SVCCA

The input to SVCCA [34] are two matrices, $\mathbf{H}_1 \in \mathbb{R}^{P \times N_1}$ and $\mathbf{H}_2 \in \mathbb{R}^{P \times N_2}$, which hold the state vector representations of two RNNs over $P$ test inputs. Here, $N_1$ and $N_2$ denote the number of neurons in each RNN (in general $N_1 \neq N_2$). First, the singular value decomposition (SVD) is computed for each matrix: $\mathbf{H}_1 = \mathbf{U}_1 \mathbf{S}_1 \mathbf{V}_1^\mathsf{T}$ and $\mathbf{H}_2 = \mathbf{U}_2 \mathbf{S}_2 \mathbf{V}_2^\mathsf{T}$. Then, these decompositions are truncated by taking the top $R$ singular vectors. The value of $R$ is a user-defined hyperparameter. Let $\widetilde{\mathbf{V}}_1 \in \mathbb{R}^{R \times N_1}$ and $\widetilde{\mathbf{V}}_2 \in \mathbb{R}^{R \times N_2}$ denote the truncated right singular vectors. Finally, canonical correlations analysis (CCA; [57]) is performed to quantify the similarity of $\widetilde{\mathbf{V}}_1$ and $\widetilde{\mathbf{V}}_2$.

## E  Centered kernel alignment (CKA)

Centered kernel alignment (CKA; [35]) is a measure of similarity between representations that is invariant to orthogonal transformation and isotropic scaling, but unlike SVCCA, is not invariant to invertible linear transformations. It defines a similarity between two representations $X \in \mathbb{R}^{m \times n_x}$ and $Y \in \mathbb{R}^{m \times n_y}$ where $m$ is the number of examples, and $n_x$ and $n_y$ are the number of units in the representations for $X$ and $Y$, respectively. The measure can be computed (for a linear kernel, see [35] for details) as:

$$\mathrm{CKA}(X, Y) = \frac{\|X^T Y\|_F^2}{\|X^T X\|_F \|Y^T Y\|_F}$$

Below, we compare representations using both singular vector canonical correlation analysis (SVCCA) and centered kernel alignment (CKA). Each figure shows comparisons for each of the three studied tasks. Within each figure, the first row shows the full pairwise distance matrix using either SVCCA (left column) or CKA (right column). The next two rows show embeddings of a subset of these pairwise distances in 2D using multi-dimensional scaling, highlighting differences by architecture (middle row) or activation (bottom row). The key takeaway is that both SVCCA and CKA show differences between network representations that cluster based on RNN architectures.

Figure 6: Comparing SVCCA and CKA for the flip flop task. See Appendix E for description of the panels.

Figure 7: Comparing SVCCA and CKA for the sinewave task. See Appendix E for description of the panels.

Context dependent integration (CDI) task

SV-CCA          Linear CKA

Vanilla
LSTM
GRU
UGRNN

relu
tanh

Figure 8: Comparing SVCCA and CKA for the context dependent integration task. See Appendix E for description of the panels.