[Reviews · NeurIPS 2019]

Reviewer 1



UPDATE after rebuttal: authors have addressed some of my concerns, so I'm updating my score to 8. To summarize, this paper aims to shed light on the connections between artificial recurrent neural networks and biological networks, in order to gain insight into neural circuit functionality through studying RNNs. More specifically, the paper comments on the ability for RNNs to mimic the behavior SNNs and neural recordings despite a vast difference in their inherent architectures. Such a phenomenon may suggest neural invariants, which act universally (in the context of a task) across either all RNN and SNN architectures, or broader groups containing various architectures in each. The paper does not look at neural recordings or SNNs, but instead trains 96 RNNs of various combinations of architectures, activations, network sizes, and L2 regularizations on three separate tasks (discrete memory, pattern formation, and analog memory) common to computational neuroscience. For each task singular value canonical correlation analysis (SVCCA) and MDS are used to determine the representational geometry of the RNNs and a numerical approach to dynamical systems analysis (and again with MDS) is used to gain insight into the topological stability structure. Three low-dimensional tasks are used: the 3-bit flip-flop is 3D, sine wave is 2D, and CDI is 2D. Given that these intrinsic structure of the problems, it is not surprising that the trained RNNs find low-dimensional solutions. Details about the training accuracy is not provided (please do). I'm assuming all RNNs achieved near 100% performance? The effectiveness of analysis (SVCCA & fixed point graph) are only presented visually using MDS. Although this is convincing in most cases, it is qualitative only. I suggest computing some sort of similarity metric that summarizes these results. (not suggesting any hypothesis test) line 193: MDS is very weak support of this claim Since Vanila is a special case of all the rest (which should be mentioned), any difference found indicates non-trivial difference in the solutions. Yet, the analysis suggests that there is little difference except in CDI. If Vanila can solve the task (I don't know since the performance is not reported in the paper), these differences are due to which solutions are easier to train from the initial conditions, right? Does the gated RNNs diverge from the tanh RNN solution if initialized close by? For the sine wave task, was there always just 1 fixed point per fixed input? If that's not the case, how would one create a topology of fixed points? In theory, the RNN can find solutions without input dependent fixed point, right? Also, the key feature here is the limit cycle, not the fixed point. This seems to be a major limitation of this method. Fig 2f does not mean much to me. Am I missing something? Why is this interesting? line 212: what is the implication of Fig 2e? For CDI analysis, a line attractor is presumed for the analysis. If there was always one fixed point per condition, it is very likely that every network would give identical "topology" due to the task structure, not due to what the network learned. Fig 1c needs colorbar Fig 1d at first glance I couldn't tell that these are two different MDS visualizations Fig 1e fixed point graph is too small to see line 86: Supplement line 156: Fig. 1b should be 1c line 238: "These results show that the all", line 220: "of architecture (right) and activation (left)" these should be reversed

Reviewer 2



POST REBUTTAL UPDATE: I thank the reviewers for their detailed rebuttal. I agree with the argument for CCA usage. Artificial neural networks – both feedforward and recurrent – are increasingly being used in neuroscience as hypothesis generators. That is, networks are trained on cognitive-like tasks, and then aspects of the artificial network are compared to biology. Despite the increasing usage of this approach, there are very few systematic efforts to understand the “rules of the game”. The present contribution is thus a very timely systematic investigation of which aspects of trained RNNs are variant and which are invariant. Specifically, the authors compare a “tensor” of (architecture * task * analysis method). They conclude that network geometry, as assessed by CCA is highly architecture dependent. In contrast, fixed point topology is mostly task dependent. Major comments: 1. The tasks used seem to have only one possible solution strategy, and thus the topology result could be somewhat trivial. Is that indeed the case? Can different tasks break this? 2. CCA assumes linearity. RNNs are nonlinear. Did you try a nonlinear method? Minor comments: 3. Related work and line 60-61 “theoretical clarity… completely lacking”. While there is very little theory, there is some that could be relevant [1]–[3]. 4. Page footer is NeurIPS 2018 5. K bit flip-flop. Input statistics (rate) are not specified 6. MDS graphs (e.g. figure 1D) : Perhaps I’m missing something, but I expected to see the same distribution of points colored differently in the left and right parts of the panel. The text says that you used a similarity matrix between all networks to construct the MDS space, and then projected to 2D. This should give a scatter of uncolored points, that you can later color according to tanh/relu or architecture. 7. Figure 1C : axes labels, colorbar – either on the figure, or in the caption. Are the networks ordered according to clustering? According to MDS? 8. Figure 1E the graph looks undirected, which is misleading. 9. Line 211, Figure 2F : “systematic differences” : Are there error bars? Is this reproducible? If so, can you hypothesize on mechanisms or implications? 10. Line 238 “that THE all” 11. Figure 4 : Was this effect similar for all configurations (task,architecture,unit type)? Or just relu/tanh in the contextual integration task? [1] A. Rivkind and O. Barak, “Local Dynamics in Trained Recurrent Neural Networks,” Phys. Rev. Lett., vol. 118, no. 25, p. 258101, Jun. 2017. [2] F. Mastrogiuseppe and S. Ostojic, “Linking Connectivity, Dynamics, and Computations in Low-Rank Recurrent Neural Networks,” Neuron, vol. 99, no. 3, pp. 609-623.e29, Aug. 2018. [3] F. Mastrogiuseppe and S. Ostojic, “A Geometrical Analysis of Global Stability in Trained Feedback Networks,” Neural Comput., vol. 31, no. 6, pp. 1139–1182, Apr. 2019.

Reviewer 3



UPDATE AFTER REBUTTAL: I want to thank the authors for the rebuttal letter and for their willingness to extend the discussion motivating the choice of tasks in terms of the computational primitives that they expose. - Originality: This type of systematic investigation is sorely needed in the literature of neural networks and RNNs in particular, despite not being particularly original in strict sense of the term - Quality: The methodology seems solid and nicely takes into account quantitative methods previously developed in the literature - Clarity: The paper is clear and well-written. The motivation is very convincingly laid out. - Significance: Beside being significant for its results, the paper is significant in that it exemplifies a type of scientific approach that is sorely missing in the literature.

[Author Response · NeurIPS 2019]

We wish to thank all of the reviewers for their time and for their thorough reading of our paper! All minor corrections and typos have been addressed, major concerns/comments are addressed below:

**Reviewer #2** We have added text in the discussion about limitations of the fixed point analysis. Other comments addressed in order: **(1)** The reviewer comments that "it is not surprising that the trained RNNs find low-dimensional solutions." We agree! We verified that we never described this fact as 'surprising' in the text. **(2)** All RNNs do indeed achieve close to zero error. Figure 1 (below) has been added to the supplement, it shows the histogram of mean squared error across all networks for each task.

**(3)** We compare SVCCA and fixed point analyses visually using MDS as it is an intuitive visualization of the corresponding distance matrices (e.g. in Fig. 1c). For any scientific conclusions that necessitate quantitative comparisons, the distance matrix can be used (we have elaborated this point in the text). **(4)** The vanilla RNN is a kind of special case of the

Figure 1: Final losses across all studied networks.

gated networks, in that the weights of the gated architectures may be initialized such that the resulting network is effectively a vanilla RNN. After initializing the weights a gated architecture at the solution of a vanilla RNN, we suspect that the weights will not change as the vanilla RNN solves these tasks perfectly well. Indeed, gated architectures were introduced to help with trainability in recurrent networks; rather than increasing the expressivitiy or capacity of these models. **(5)** For the sine wave task, there was always a unique fixed point per fixed input. In general, a limitiation of the topology analysis is that it the same number of fixed points between models (we now elaborate on this limitation in the discussion). Regarding the "key feature being the limit cycle", that is the point of Fig. 2f (see our next response). **(6)** Fig. 2f is an attempt to quantify the limit cycle. It demonstrates that the linearized dynamical system is a good approximation of the full nonlinear dynamics, in that the predicted linear frequency (estimated using the top eigenvalues of the Jacobian of the system) matches the task frequency. This is an important validation of why we should trust the linearization analysis, as such, we think its inclusion is warranted. **(7)** We have added a sentence with our interpretation of Fig 2e after Line 212: the implication is that every network has a 1D manifold of input-dependent fixed points. **(8)** For CDI analysis, we do not presume that there is a line attractor, rather, the topological analysis reveals the presence of the line attractor across networks.

**Reviewer #3** **(Major point 1.)** Our analysis has revealed that nearly all of the networks use similar strategies to solve the task, but this was not a foregone conclusion when we began our study. Moreover, we occasionally find networks (typically with extreme values of l2 regularization) with large amplitude, fast oscillations on top of the existing solution (e.g. they still have a line attractor in CDI). These have qualitatively different trajectories, but seem pathological as the these oscillations have no projection on the readout. Note that Sussillo et. al. 2015 similarly reported chaotic solutions under large weight initialization. We are including a depiction of these fast oscillatory networks in the supplement for completeness, along with corresponding discussion in the main text. **(Major point 2.)** Although CCA compares linear projections of two representations, we believe this is a necessary restriction. Under arbitrary nonlinear transformations, any two representations can be made to look identical. Moreover, our motivation in this work is to better understand the surprising similarities found between artificial and biological networks, which use measures such as CCA. Under this motivation, we used CCA in order to be able to compare to the existing literature. In addition, fixed point topology speaks to representation as well and yields identical results for nonlinearly warped, but otherwise similar topologies. **(Minor comment 3.)** We appreciate the pointers to relevant work! We have added these citations to the text. **(Minor comment 4.)** For the MDS graphs, for visualization purposes, we filtered the distance matrix to highlight differences due to architecture (left; formed by taking only tanh networks) or nonlinearity (right; formed by taking only vanilla RNNs). We have generated the requested MDS visualization (comparing all networks) for each task; these will be added in the supplement—the central results do not change. **(Minor comment 5.)** Yes, there are error bars (shaded patches in Fig. 2F); thus it is reproducible. We have not identified the reason for these differences, presumably they arise due to differences in higher order terms of the RNNs. **(Minor comment 11.)** We only ran the analysis for the relu/tanh CDI networks, as the purpose of the figure is just to point out how SVCCA may be misleading.

**Reviewer #4** **(1)** Although we would love to develop theory for understanding universality in recurrent networks, we believe that empirical evidence for universality in RNNs is a first step to a broader theoretical understanding. Indeed, a number of studies are beginning to make progress on this front (which are now cited in the main text), but with some simplifying assumptions on the tasks or networks being studied. Given the large diversity of tasks and networks that are actively used in the literature, we feel that the empirical approach taken in this paper (which allows us to explore a large set of tasks/architectures) is a significant contribution on its own. **(2)** We have added a deeper discussion regarding the chosen tasks in both the introduction (motivating why we chose them) and discussion (where we discuss limitations of these particular tasks). We think this particular set of tasks is interesting because they cover fundamental computational primitives: discrete and analog memory, pattern generation, and contextual computation.

[Meta-Review · NeurIPS 2019]

Dear authors, congrats on the acceptance-- please do take the feedback in the reviews into account when preparing your final manuscript.